# Fast generation of Schrödinger cat states using a Kerr-tunable superconducting resonator

X. L. He [1,2,7], Yong Lu [3,4,7] ✉, D. Q. Bao[1,2], Hang Xue [1,2], W. B. Jiang[1,2], Z. Wang[1,2], A. F. Roudsari [4], Per Delsing [4], J. S. Tsai[5,6] & Z. R. Lin [1,2] ✉

Schrödinger cat states, quantum superpositions of macroscopically distinct classical states, are an important resource for quantum communication, quantum metrology and quantum computation. Especially, cat states in a phase space protected against phase-flip errors can be used as a logical qubit. However, cat states, normally generated in three-dimensional cavities and/or strong multi-photon drives, are facing the challenges of scalability and controllability. Here, we present a strategy to generate and preserve cat states in a coplanar superconducting circuit by the fast modulation of Kerr nonlinearity. At the Kerr-free work point, our cat states are passively preserved due to the vanishing Kerr effect. We are able to prepare a 2-component cat state in our chip-based device with a fidelity reaching 89.1% under a 96 ns gate time. Our scheme shows an excellent route to constructing a chip-based bosonic quantum processor.

Quantum computation has been proven to surpass classical architectures in certain computational tasks[1]. Quantum information has been encoded and manipulated in diverse systems such as cold atoms[2], trapped ions[3,4], superconducting circuits[5]. Especially, superconducting circuit is a promising platform which has shown significant progress on the gate-based quantum computers[1,6]. Additional qubit elements are normally required to achieve large-scale error-correctable two-level system-based quantum computation[7]. In contrast, the phase space of a bosonic system inherently provides a larger Hilbert space and thus a larger coding area[8–11]. Therefore, encoding quantum information in continuous variables leads to a significant reduction in hardware overhead on the path towards the fault-tolerance[12,13]. The nonclassical states with negative Wigner functions[14,15] can be regarded as a quantum computing resource to obtain quantum computational advantage. Recently, non-classical states including Schrödinger's cat codes[16] binomial codes[17], GKP states[18,19], and cubic-phase states[18,20],

have been demonstrated in cavities coupled to ancillary qubits. However, the ancillary qubit normally has a fixed Kerr nonlinearity which might be detrimental even for the storage of nonclassical state[21]. In previous results, Schrödinger's cat states were mostly generated by engineering the two-photon losses[22,23] or ancilla-assisted processes[24] in two-[25,26] and three-dimensional[23] structures.

In this paper, differently from traditional gate-based cavity control schemes using the dispersive shift of a nominally linear resonator to an ancilla qubit[24–26], our cat state preparation scheme is an alternate by applying a displacement followed by a Kerr gate to a nonlinear resonator. The Kerr gate is implemented by quickly tuning the nonlinearity of the resonator terminated by a Superconducting Nonlinear Asymmetric Inductive eLement (SNAIL)[27,28] as shown in Fig. 1a. Moreover, by tuning the flux bias to the Kerr-free point with eliminated four-wave mixing term, we therefore preserve the prepared cat states against the Kerr-induced evolution.

[1]National Key Laboratory of Materials for Integrated Circuits, Shanghai Institute of Microsystem and Information Technology, Chinese Academy of Sciences, 200050 Shanghai, China. [2]University of Chinese Academy of Science, 100049 Beijing, China. [3]3rd Physikalisches Institut, University of Stuttgart, 70569 Stuttgart, Germany. [4]Microtechnology and Nanoscience, Chalmers University of Technology, SE-412 96 Göteborg, Sweden. [5]Graduate School of Science, Tokyo University of Science, Shinjuku, Tokyo 162-0825, Japan. [6]Center for Quantum Computing, RIKEN, Wako, Saitama 351-0198, Japan. [7]These authors contributed equally: X. L. He, Yong Lu. ✉e-mail: kdluyong@outlook.com; zrlin@mail.sim.ac.cn

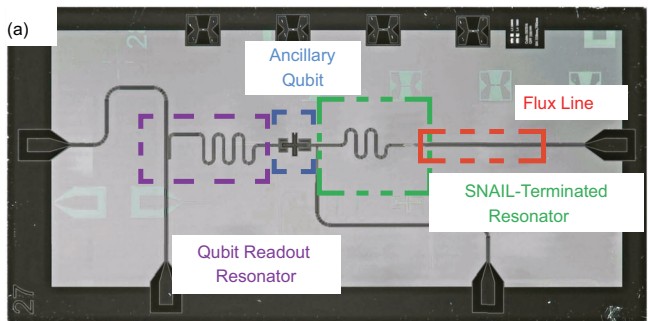

(a)

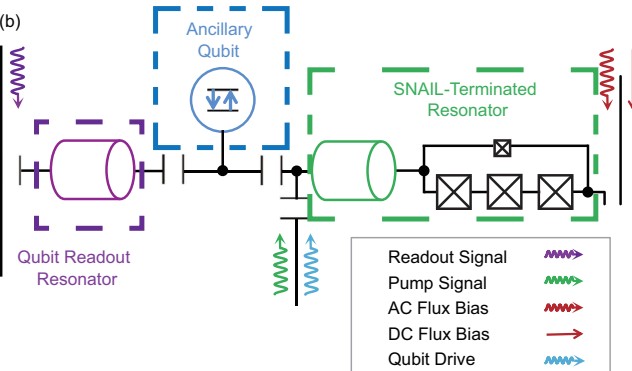

(b)

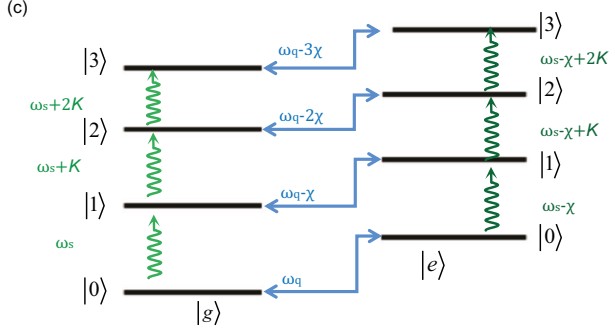

(c)

**Fig. 1 | Structure of the superconducting circuit. a** A microscopic photo of the superconducting circuit. An ancillary qubit in the middle is capacitvely coupled to both a readout resonator (left) and a SNAIL-terminated resonator (right).
**b** Schematic circuit diagram of the system. **c** Energy structure of the dispersively coupled nonlinear resonator and qubit. $|g\rangle$ and $|e\rangle$ are the ground and excited state of qubit respectively. $|0\rangle, |1\rangle, |2\rangle, \ldots$ represent the energy levels of the resonator. $\chi$ is the dispersive shift.

## Results

### Design of the quantum circuit

The energy of the SNAIL in our circuit with three big junctions and one smaller junction [Fig. 1c] can be written as[29]

$$U_{\text{SNAIL}}(\varphi) = -\beta E_J \cos(\varphi) - 3E_J \cos\left(\frac{\varphi_{\text{ext}} - \varphi}{3}\right), \quad (1)$$

where the ratio of the Josephson energies of the small and the big junctions of SNAIL, $\beta \approx 0.095$, the Josephson energy $E_J/h \approx 830$GHz[28] (Details in Methods), $\varphi_{\text{ext}} = 2\pi\Phi_{\text{ext}}/\Phi_0$ is the phase induced by the external magnetic flux and $\varphi$ is the phase difference between two ports of the SNAIL. The Hamiltonian of the SNAIL-terminated resonator is[10,27,30]:

$$H_{\text{SNAIL-Res}} = \hbar\omega_s a^\dagger a + g_3(a + a^\dagger)^3 + g_4(a + a^\dagger)^4, \quad (2)$$

where $\omega_s$ is the resonant frequency of the SNAIL-terminated resonator (the tunable range of $\omega_s/2\pi$ is around 4.08-5.00 GHz in our device). $a$

$(a^\dagger)$ is the annihilation (creation) operator. $g_3(g_4)$ is the coupling strength for the three (four)-wave mixing.

Including the coupled ancillary transmon qubit, the total effective Hamiltonian of the system in the dispersive regime is given by[31]

$$\begin{aligned} \frac{H_{\text{eff}}}{\hbar} &\approx \omega_s a^\dagger a + K a^{\dagger 2} a^2 + \frac{\omega_q}{2} b^\dagger b \\ &\quad - \frac{\chi}{2} a^\dagger a b^\dagger b - \frac{K_q}{2} b^{\dagger 2} b^2, \end{aligned} \quad (3)$$

where $\omega_q$ is the frequency of the ancillary qubit (around 5.09-5.19 GHz). $b$ $(b^\dagger)$ is the lowering (raising) operator for the ancillary qubit. $K$ is the Kerr nonlinearity of the resonator, defined as the frequency shift per photon, $K = K_s + K_{qs}$ with the self-Kerr term $K_s = 12(g_4 - 5g_3^2/\omega_s)$ from the SNAIL element and the cross-Kerr term $K_{qs} = \chi^2/4K_q$ from the qubit with the dispersive shift $\chi/2\pi \approx 3.5 - 18$MHz depending on the flux bias [especially $\chi/2\pi \approx 4.35$MHz when the external flux $\Phi_{\text{ext}} = 0.4026\Phi_0$ (see Methods)]. The qubit anharmonicity is $K_q/2\pi \approx -420$MHz. The value of $K_s/2\pi$ can be tuned from negative to positive with a range up to a few MHz [Fig. 2a], whereas the value of $K_{qs}/2\pi$ is always negative on the order of kHz. Therefore, it is possible to cancel the cross-Kerr term from the qubit to obtain $K = 0$ by tuning $K_s$ with the magnetic flux through the SNAIL[32], see details in Table 1.

### Characterization and manipulation of the nonlinearity

Firstly, we calibrate the values of the Kerr coefficient $K$ [Fig. 2a] precisely with two approaches, namely single-tone and two-tone measurements [Fig. 2b]. For the single-tone measurement, we sweep the frequency of a displacement pulse $D(\alpha)$ followed by a conditional qubit $\pi$-pulse, where the qubit is excited only if the SNAIL-terminated resonator is empty ($\alpha$ is the displacement with photon number $N = \alpha^2$). Therefore, we can observe the resonator frequency shift with the photon number inside as shown in Fig. 2c–e. We can extract the Kerr coefficient $K$ by linearly fitting the relationship between the frequency shift and the photon number (see Methods). This method is valid only for a small $K$ so that the total frequency shift is not larger than the pulse linewidth. For a larger $K$, we switch to perform a two-tone measurement. In this measurement, we regard the nonlinear resonator as a multi-level system with an anharmonicity (similar to a qubit), where we perform Rabi oscillations on the lowest three levels by applying two pulses on the transition $|0\rangle \Rightarrow |1\rangle$ and $|1\rangle \Rightarrow |2\rangle$, respectively. Thus, the anharmonicity, corresponding to the value of $K$, can be obtained as soon as the resonant frequencies are found (see Methods).

As shown in Fig. 2a, the dynamic range of the Kerr coefficient $K/2\pi$ is approximately from −5 MHz to 6 MHz which is close to the theoretically simulated result[29]. Particularly, with flux bias $\Phi_{\text{ext}} = 0.4026\Phi_0$, we find a working point where $K$ is small, $|K/2\pi| < 70$ kHz from single-tone measurement. The accuracy is limited by the spectroscopic linewidth of the pump pulse. Therefore, the Kerr-induced dynamic evolution is negligible within a time scale on the order of microseconds. At the more accurate Kerr-free point, the nonlinearity from the qubit $K_{qs}/2\pi = \chi^2/4K_q \approx -11$ kHz can be compensated by $K_s$, where the dynamic evolution is ideally eliminated. In order to show the merits of preparing the quantum state at the Kerr-free point, as an example, we displace the nonlinear resonator with $\alpha = 1.42$ at $\Phi_{\text{ext}} = 0.4026\Phi_0(K \approx 0)$ and $\Phi_{\text{ext}} = 0.41\Phi_0(K/2\pi \approx 0.5$ MHz), respectively. Then, we wait for a time duration $\Delta t$ before performing the Wigner tomography on the states by taking the parity measurements[33,34], where the pulse sequence is shown in Fig. 3a. The results [Fig. 3b–g] clearly illustrate that the quantum states can be preserved well at Kerr-free point whereas the phase of the state collapses when the Kerr nonlinearity is nonzero. Moreover, the frequency shift among energy levels may induce variations in the photon distribution (as what we discussed in the two-tone nonlinearity measurement above). Injection of multiple photons would be much easier

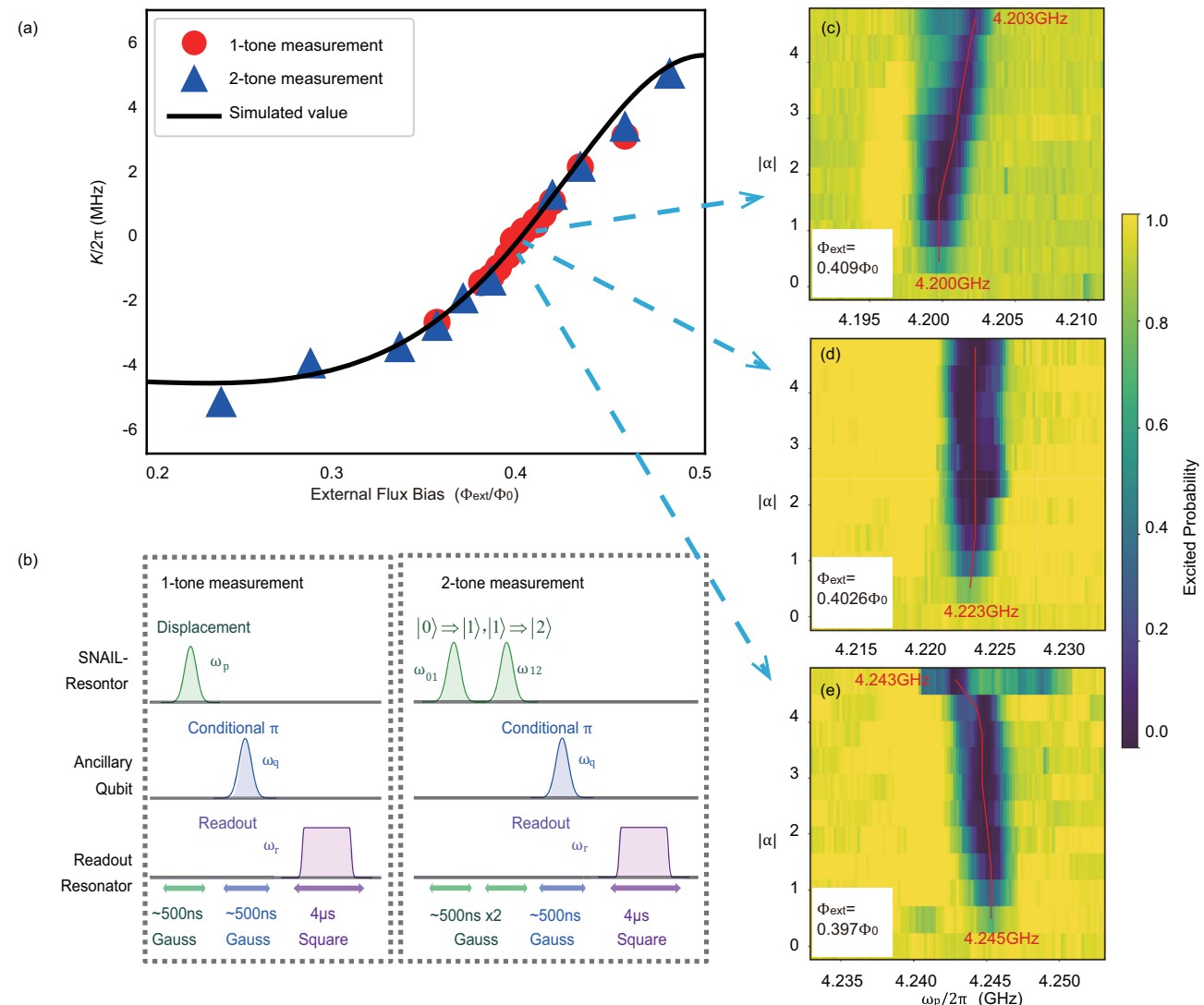

**Fig. 2 | Experimental methods to calibrate the tunable nonlinearity of the resonator. a** Flux bias dependent nonlinearity. The Kerr coefficient $K$ is measured with single-tone and two-tone measurements. **b** Pulse sequences for the single-tone and two-tone nonlinearity measurements. **c–e** Results of the single-tone measurement near the Kerr-free point. $\alpha$ is the displacement. $\omega_p$ is the frequency of the pump pulse to the SNAIL-terminated resonator.

at the Kerr-free point because of the simple spectrum[28]. As a result, a larger Hilbert space of photons provides us with a larger coding area either for error correction[9,35,36] or loss suppression[26].

**Preparation of Schrödinger cat states**

Furthermore, the fast tunablity of the Kerr coefficient can also be used to generate non-classical quantum states. The Kerr cat qubits[23] with the related error correction methods[37] benefiting from Kerr nonlinearity show the potential of dissipation-insensitive and long lifetime quantum computation in a multidimensional Hilbert space.

**Table 1 | Parameters of the circuit. ($\omega_{qO}$, $\omega_{sO}$, and $\chi_O$ are the values at Kerr-free working point.)**

| $\omega_q/2\pi$ | 5.09–5.19 GHz | $\omega_s/2\pi$ | 4.08–5.00 GHz |
|---|---|---|---|
| $K_q/2\pi$ | −420 MHz | $K_s/2\pi$ | (−5) −6 MHz |
| $E_J/h$ | 830 GHz | $\beta$ | 0.095 |
| $\chi/2\pi$ | 3.5–18 MHz | $\omega_{qO}/2\pi$ | 5.095 GHz |
| $\omega_{sO}/2\pi$ | 4.223 GHz | $\chi_O/2\pi$ | 4.35 MHz |

For example, we consider the Bloch sphere of a Kerr-cat qubit which is constructed with a group of perpendicular states[23]:

$$\left|\varphi_{\pm X}\right\rangle = |\pm\alpha\rangle, \tag{4}$$

$$\left|\varphi_{\pm Z}\right\rangle = |\alpha\rangle \pm |-\alpha\rangle, \tag{5}$$

$$\left|\varphi_{\pm Y}\right\rangle = |\alpha\rangle \pm i|-\alpha\rangle, \tag{6}$$

$\left|\varphi_{\pm X}\right\rangle = |\pm\alpha\rangle$ are the coherent states generated by pumping our nonlinear resonator with coherent pulses. To prepare the cat states $\left|\varphi_{\pm Y}\right\rangle = |\alpha\rangle \pm i|-\alpha\rangle$, the Kerr nonlinearity normally plays an important role[23,38]. As shown in Fig. 4a, the flux bias pulse (with a pulse width $\tau$) after the first displacement $D(\alpha)$ introduces a flux bias shift, as well as a large Kerr coefficient $K$. The coherent states evolution under this nonlinear Hamiltonian results in phase shifts among Fock states $|N\rangle$ ($N = 0, 1, 2, \ldots$), which means the rotating speed in the phase space is not uniform. Therefore, when we initialize the system with a coherent state $|\alpha\rangle$, the evolution of the field states (during the flux bias pulse in

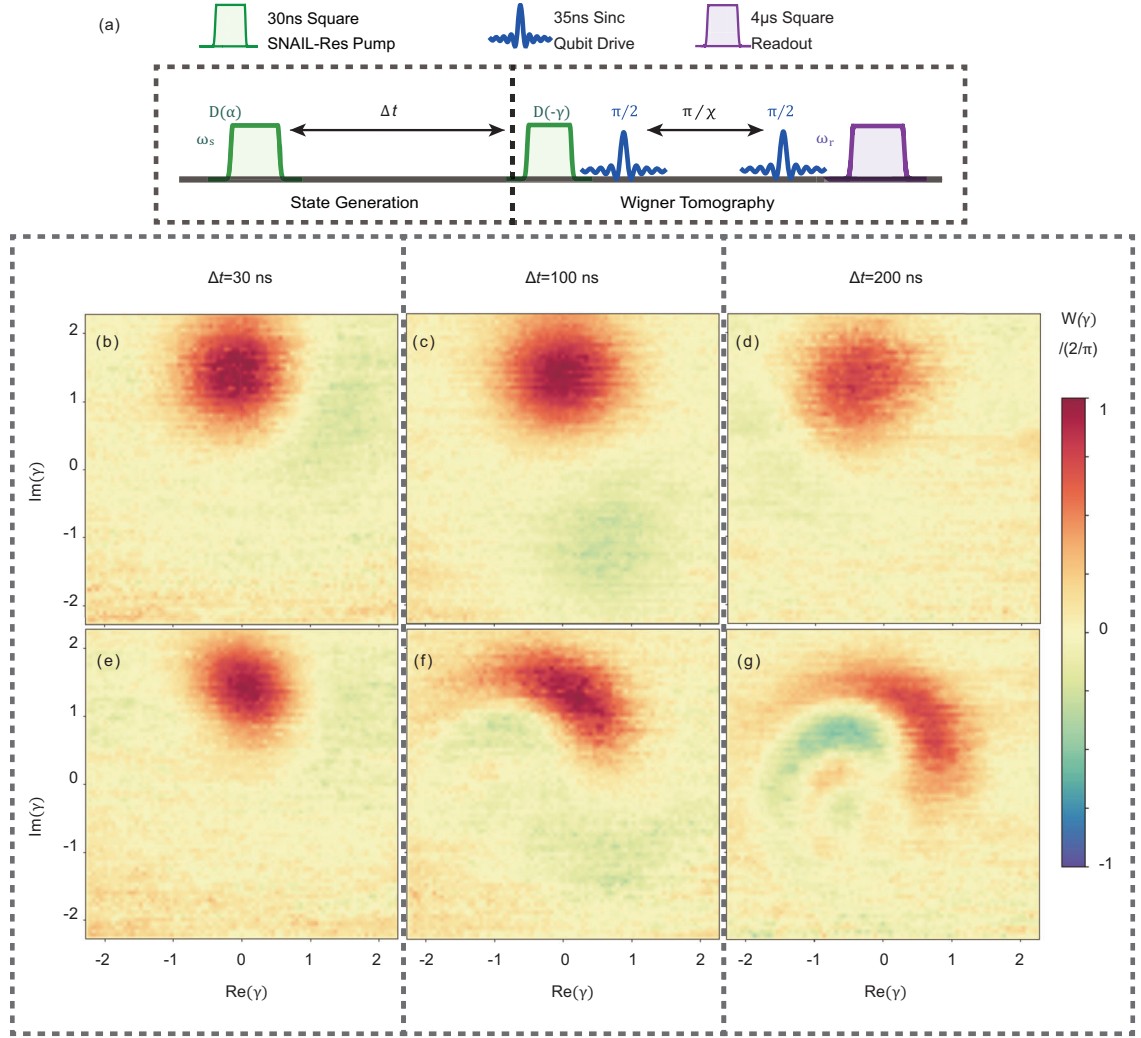

**Fig. 3 | Time evolution of a coherent state in the SNAIL-terminated resonator. a** The pulse sequence where the pulse width of displacement is 30ns, and the time between two displacement pulses is $\Delta t$. The experimental Wigner tomography shows the evolution progress of the coherent states at different time durations as shown in **b–d** at the Kerr-free point and **e–g** with $K/2\pi \approx 0.5$MHz where the Kerr effect clearly distorts the state.

Fig. 4a) can be written as[23]

$$\left|\Psi(\tau)\right\rangle = e^{i\frac{K}{2}(a^\dagger a)^2 \tau}\left|\alpha\right\rangle$$
$$= e^{-|\alpha|^2/2}\sum_N \frac{\alpha^N}{\sqrt{N!}}e^{i\frac{K}{2}N^2\tau}\left|N\right\rangle \tag{7}$$

Under certain circumstances, the $m$-component cat states can be generated when the nonlinear evolution time $\tau = \tau_0/m$ with $\tau_0 = 2\pi/K$ ($m = 1, 2, \ldots$). In particular, when

$$\tau = \pi/K \quad \text{or} \quad 3\pi/K, \tag{8}$$

the final state is

$$\left|\Psi(\tau)\right\rangle = \left|\alpha\right\rangle \pm i\left|-\alpha\right\rangle. \tag{9}$$

In our experiment, we chose $\alpha = 1.42$ and a flux bias with $K/2\pi = 5.21$ MHz corresponding to $\tau_0 = 192$ ns. When the pulse width of the flux bias is either $\tau = \tau_0/2 = 96$ns or $\tau = 3\tau_0/2 = 288$ ns, we get $\left|\varphi_{\pm Y}\right\rangle = \left|\alpha\right\rangle \pm i\left|-\alpha\right\rangle$. By setting $\tau = \tau_0/3$ (64 ns) and $\tau = \tau_0/4$ (48 ns), we also implement 3- and 4- component cat states which can be used

for quantum error correction[37,39]. Wigner functions of 2, 3 and 4-component cat states are measured and shown in Fig. 4b–d.

The fidelity of the above Schrödinger's cat states can be calculated by comparing the measured Wigner function $W_{\text{meas}}$ [Fig. 4b–d] with the numerical one $W_{\text{cal}}$ [Fig. 4e–g]. The fidelity can be written as[40]

$$F = \pi \int d\gamma^2 W_{\text{meas}}(\gamma)W_{\text{cal}}(\gamma), \tag{10}$$

where $\gamma$ is the displacement vector, integrated through the whole phase space.

Here, the fidelity $F_m$ of the $m$-component cat from our measurements is

$$F_2 = 89.1\%, F_3 = 81.3\%, F_4 = 83.15\%. \tag{11}$$

Note that the distortion caused by the imperfect Wigner tomography has not been eliminated. Moreover, the fidelities are currently limited by our device coherent time 1μs (see Supplementary Fig. 1). The dominating dephasing source of cat states is the single photon loss in our system (details in Supplementary Fig. 3). Additionally, we need to mention that the ancillary qubit is a nonnegligible source of the collapse and decoherence of the bosonic quantum states. It is therefore

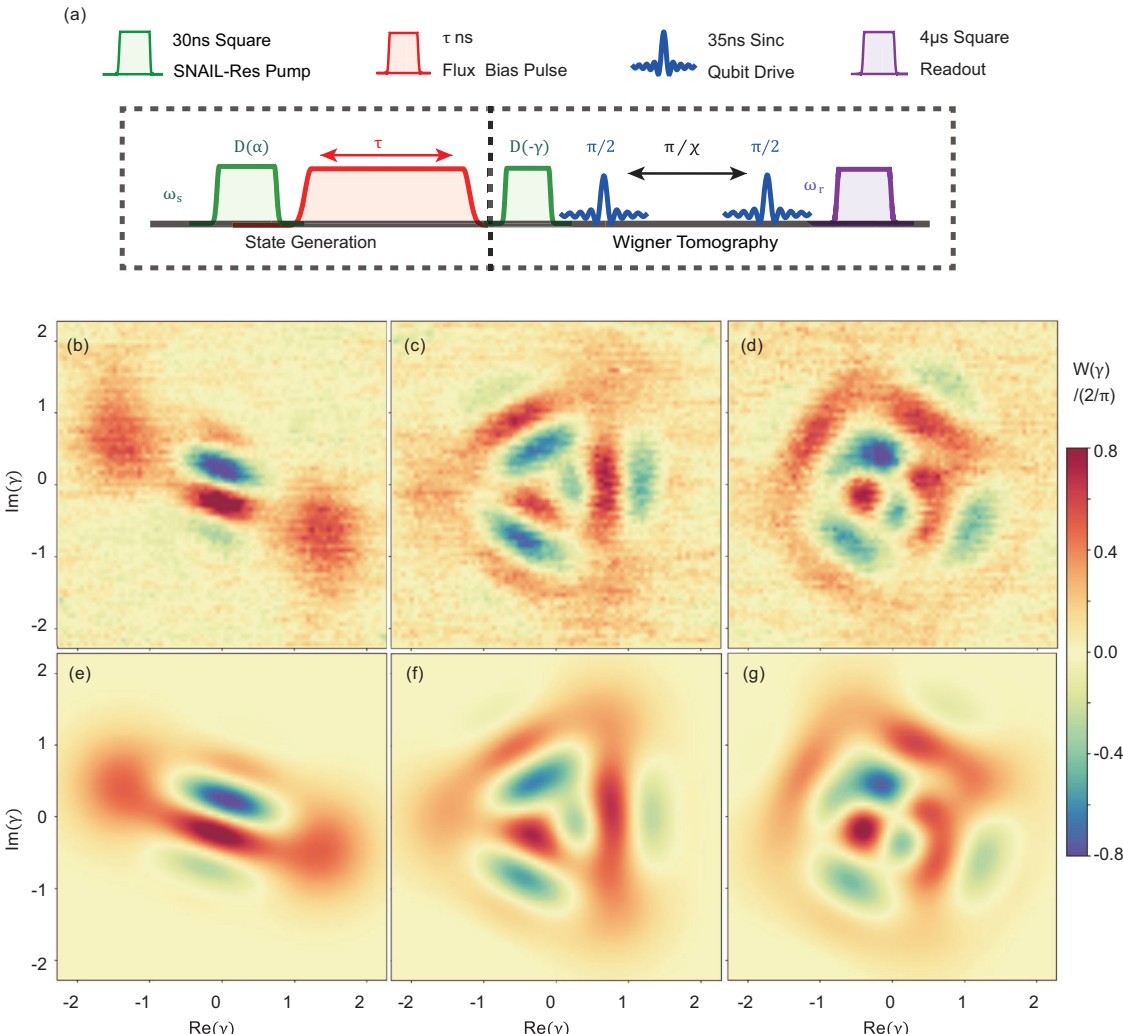

**Fig. 4 | Schrödinger cat states generation through the fast modulation of Kerr nonlinearity. a** Pulse sequence for generating the *m*-component Poisson distributed cat states. **b–d** Measured Wigner functions of the *m*-component cat ($m = 2,3,4$). **e–g** Numerical Wigner functions of the *m*-component cat ($m = 2,3,4$) obtained through QuTip[48].

beneficial to reduce the photon loss and the impacts from the qubit (e.g. spectrally isolating the ancillary qubit while not in use).

In previous strategies with a fixed Kerr coefficient[23,38], the *m*-component cat state is stabilized by applying a squeezing drive continuously. However, in our case, the cat states can be maintained in the Kerr-free system passively. Therefore, after the state generation, the system is immediately tuned back to the Kerr-free point, where our resonator can be described by a linear Hamiltonian[41], leading to a better storage and evolution of the cat states within a desirable lifetime (see Methods).

To verify the feasible controllability of our nonlinear resonator, we successfully generate an odd cat state, $|\varphi_{-Z}\rangle = |\alpha\rangle - |-\alpha\rangle$ by following the ancilla-assisted cat preparation method[42,43]. By using the spectral selectivity and different evolution induced by the dispersive shift, the odd cat state is obtained (details see Supplementary Figure 1). It shows the possibility of constructing a logical qubit with our platform. In conclusion, we have generated nonclassical states through the fast tunable nonlinearity on a SNAIL-terminated resonator where the tunable range is up to 10 MHz. Compared to the cat states in 3D cavities[44] where the state preparation is based on the ancillary qubit, our method is more straightforward from the fast tuning of the Kerr coefficient of the nonlinear resonator itself. Thus, our scheme is much simpler and has no affect from the imperfect

preparation on the ancillary qubit. Moreover, compared to the two-photon driving strategy[23,25,38], the time to prepare the cat states is about $1/K$ in our method, which is several times faster than the adiabatic case[38]. Meanwhile, by eliminating the Kerr-induced evolution, the states of light can be stored passively without consecutive pump at the Kerr-free point. Finally, our platform is more compact compared to 3D cavities, and shows the capability to integrate more elements. Therefore, our method shows a possibility of the extensible and low-crosstalk bosonic-based quantum computation in the future.

Our method provides an avenue to achieve continuous-variable quantum information processing. It can be used to achieve universal control of bosonic codes. One direct application of our circuitry is constructing hardware-efficient, loss-tolerable quantum computers with error correction codes[9,44]. Furthermore, networks of coupled resonators can be used to achieve quantum annealing architectures[45] and quantum simulations such as phase transitions[46], Gaussian boson sampling[47], etc.

## Methods

### Photon number calibration

The photon number in the resonator is measured by the spectroscopy of the ancillary qubit. Here, we pump the resonator with a coherent pulse $D(\alpha)$. As shown in Fig. 5a, due to the dispersive coupling to the qubit, the qubit frequency has a Poisson distribution related to the

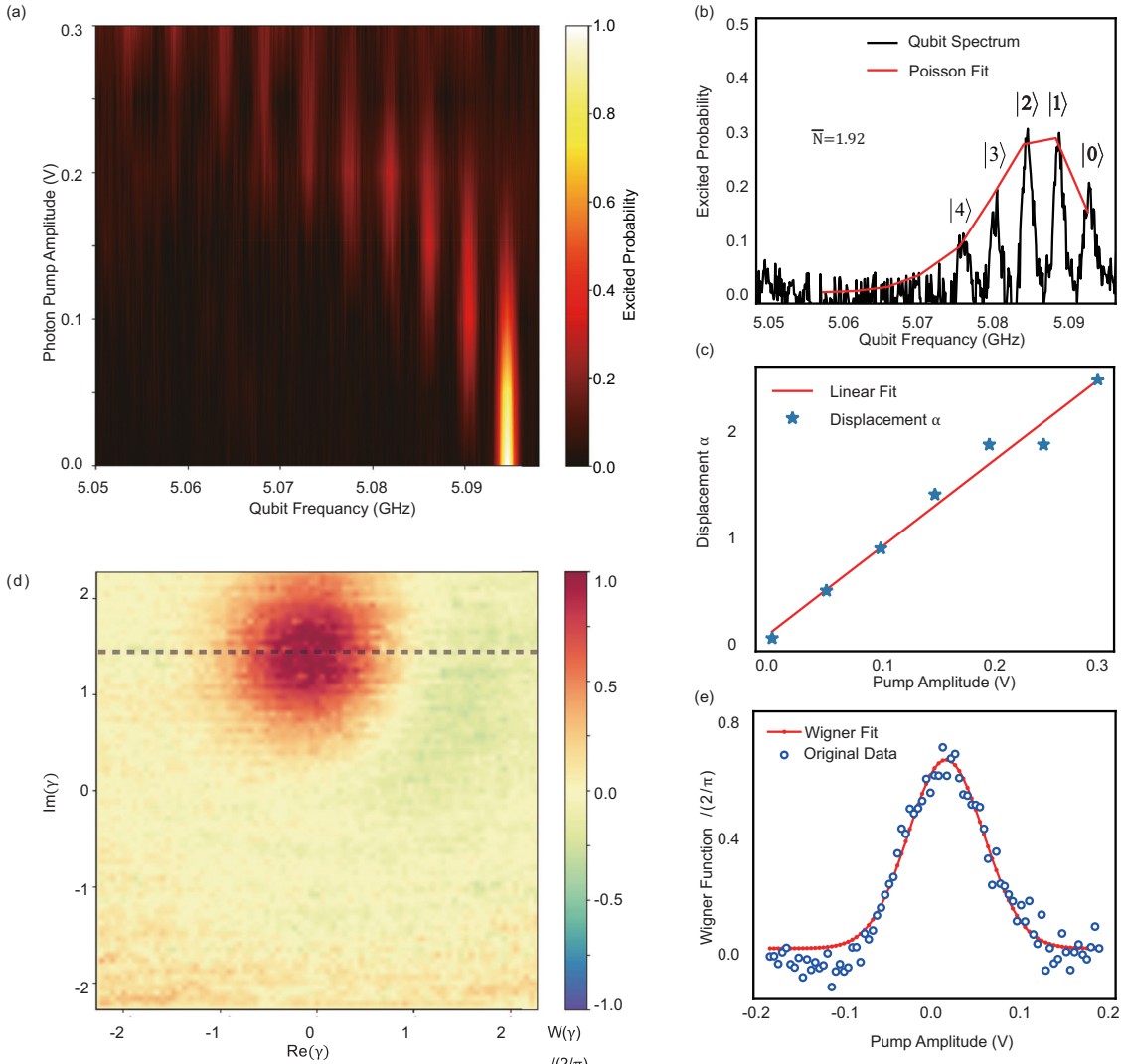

**Fig. 5 | Two strategies for photon number calibration. a** Qubit spectroscopy under different pump pulse amplitudes. **b** Poisson fitting with an average photon number 1.92. **c** Linear relationship between displacement and pump amplitude. **d** Wigner function of a coherent state with $\alpha = 1.42$. **e** Fitting of Wigner function (dashed gray line in (d)).

photon pump amplitude $V$, where the $N^{\text{th}}$ peak (away from the native qubit frequency) corresponds to the probability of the Fock state $|N\rangle$ with photon number $N$. By fitting the multi-peak spectra to a Poisson-distribution function, we can extract the average photon number $\overline{N} = \alpha^2$. Therefore, we can figure out the linear relationship between the photon pump amplitude $V$ and the values of $\alpha$, namely, $\alpha = G \cdot V$, where $G$ is the scale factor between displacement $\alpha$ and amplitude $V$.

In addition, to calibrate the photon more precisely under a low photon number ($\overline{N} < 5$), we can also get the relationship $\alpha = G \cdot V$ by fitting the Wigner distribution of the coherent state with [Fig. 5e]:

$$
\begin{aligned}
W &= \frac{2}{\pi} \exp(-2(\alpha - \alpha_0)^2) \\
&= \frac{2}{\pi} \exp(-2(G \cdot V - \alpha_0)^2),
\end{aligned}
\tag{12}
$$

where $\alpha_0$ is the initial displacement.

These relationships are used for the nonlinearity, Wigner function and lifetime measurements discussed in the main text. However, because of the flux-dependent nonlinearity, the photon-number distribution does not satisfy Poisson function when the frequency shift $N \cdot K$ is comparable with the linewidth of the pump

pulse ($\approx 2$ MHz for the 500 ns pump pulse). The photon number calibration methods above are therefore only available for $\Phi$ - $0.4\Phi_0$ with $|K/2\pi| < 2$ MHz.

## Nonlinearity characterization

The Hamiltonian of the SNAIL element with three large Josephson junctions (with Josephson energy $E_J$) and one smaller junction ($\beta E_J$) can be written as (same as Eq. (1)):

$$
U_{\text{SNAIL}}(\varphi_s) = -\beta E_J \cos(\varphi_s) - 3E_J \cos\left(\frac{\varphi_{\text{ext}} - \varphi_s}{3}\right),
\tag{13}
$$

where $\varphi_{\text{ext}}$ is the external magnetic flux induced phase through the SNAIL loop, $\varphi_s$ is the phase different between the two ports of SNAIL. Coupled with a resonator (represented by capacity C and inductance L):

$$
H = C\frac{\Phi_0^2}{2}\dot{\varphi}^2 + U(\varphi, \varphi_s),
\tag{14}
$$

$$
U(\varphi, \varphi_s) = \frac{1}{2}E_L(\varphi - \varphi_s)^2 + U_{\text{SNAIL}}(\varphi_s),
\tag{15}
$$

where $\varphi$ is the mode canonical phase coordinate and $E_L = \Phi_0^2/L$ is the inductive energy. After Taylor expansion around the minimum

point of potential $U$, the Hamiltonian of second quantization is[28,29]

$$H_{\text{SNAIL−Res}} = \hbar\omega_s a^\dagger a + g_3(a + a^\dagger)^3 + g_4(a + a^\dagger)^4,\qquad(16)$$

where

$$\hbar\omega_s = \sqrt{8E_C E_J c_2},\qquad(17)$$

$$\hbar g_3 = c_3\sqrt{E_C \hbar\omega_s}/6c_2,\qquad(18)$$

$$\hbar g_4 = c_4 E_C/12c_2,\qquad(19)$$

$$c_j = \frac{1}{E_J}\frac{d^j U}{d\varphi^j}\bigg|_{\varphi_m},\qquad(20)$$

$$E_C = e^2/2C.\qquad(21)$$

The energy level n is:

$$E_n/\hbar = n\omega_s + 6(g_4 - 5g_3^2/\omega_s)n(n+1).\qquad(22)$$

Thus, the Kerr coefficient of SNAIL-terminated resonator can be written as

$$\hbar K = \frac{d^2 E_n}{dn^2} = 12\hbar\left(g_4 - \frac{5g_3^2}{\omega_s}\right).\qquad(23)$$

The design parameters $\beta$ and $E_J$, also the relationship between injected current and flux $\varphi_{\text{ext}}$ can be extracted by fitting the flux modulation of the SNAIL-terminated resonator frequency (Fig. 6) with Eq. (22)[28].

We developed two methods for the Kerr coefficient characterization, namely, single-tone and two-tone measurements. The single-tone measurement is only suitable for smaller Kerr coefficient (e.g. $|K/2\pi| < 2\,\text{MHz}$). To contrast, larger $K$ brings more frequency shift with a similar photon number. When the frequency shift is larger than the linewidth of the single pulse, the nonlinearity can be measured by the two-tone method.

In a single-tone measurement, in order to keep the frequency sensitivity, we choose a relatively long pulse with a pulse length up to 500 ns, then we sweep the pulse frequency with following a conditional $\pi$-pulse which can excite the qubit only if the cavity is empty. Therefore, it can be regarded as a photon probe. With different pump powers, we can see the shift of resonant frequency $\Delta f$ which obeys:

$$K/2\pi = f_N - f_{N-1},\qquad(24)$$

$$K/2\pi = \Delta f/\overline{N},\qquad(25)$$

where $\overline{N}$ is the average photon number. For example, in Fig. 2c, we see the frequency shift with different displacement (i.e. pump amplitude). By linearly fitting the relationship of the frequency shift and the average photon number, we get the Kerr coefficient [Fig. 7a]. If $K$ is too large ($|K/2\pi| > 2\,\text{MHz}$), however, it is not possible to cover the frequency range with a single tone. Then, we treat the SNAIL-terminated resonator as a three level system with an anharmonicity $K$. To verify it, Rabi oscillations between energy levels ($|0\rangle \leftrightarrow |1\rangle$ and $|1\rangle \leftrightarrow |2\rangle$) are measured [Fig. 7b]. Here, the conditional qubit $\pi$ pulse is also modified to be only valid for $|0\rangle$, $|1\rangle$ or $|2\rangle$, respectively. Thus, $K$ is represented by the frequency difference of the first two pulses as

$$K/2\pi = f(|1\rangle \leftrightarrow |2\rangle) - f(|0\rangle \leftrightarrow |1\rangle).\qquad(26)$$

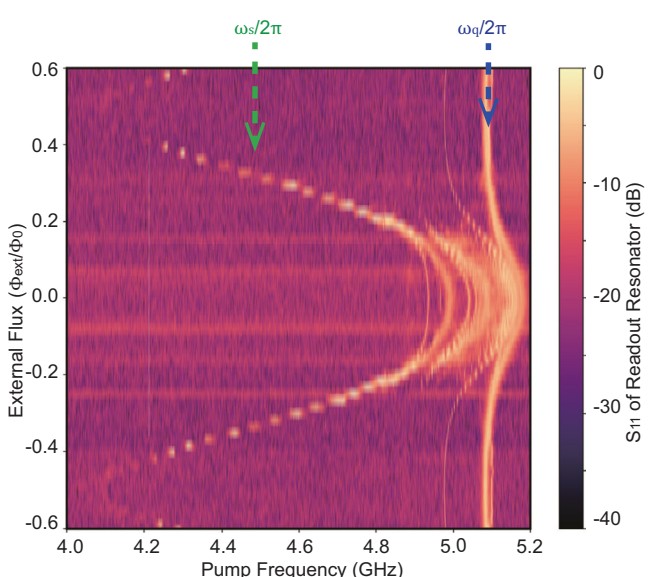

**Fig. 6 | Flux modulated spectrum.** The frequencies of the SNAIL-terminated resonator and ancillary qubit are obtained by the reflection coefficient measurement on the readout resonator with different pump frequency.

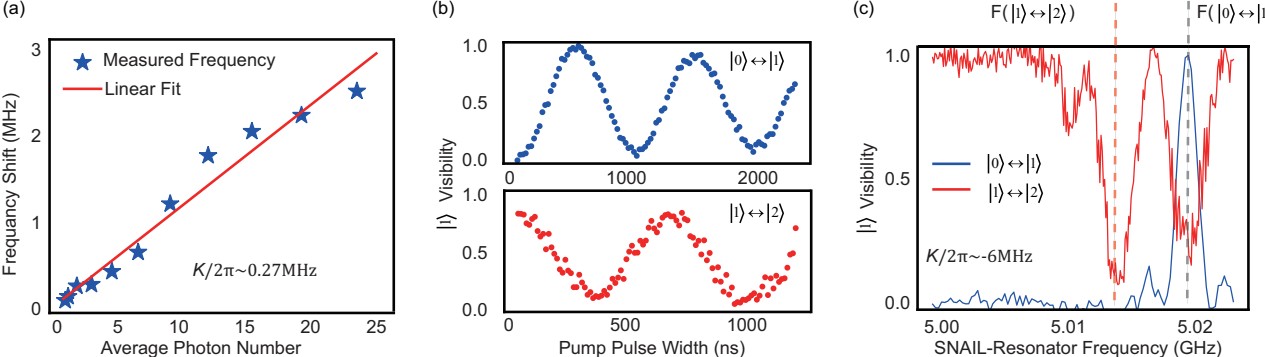

**Fig. 7 | Experimental results of 1-tone and 2-tone nonlinearity measurement.** **a** Relationship between the frequency shift and the photon number in 1-tone measurement with external flux $\Phi_{\text{ext}} - 0.409\Phi_0$. **b** Rabi oscillation between the Fock states ($|0\rangle \leftrightarrow |1\rangle$ and $|1\rangle \leftrightarrow |2\rangle$). **c** Frequency scan of the SNAIL-terminated resonator under a 2-tone measurement with a conditional $\pi$ pulse for $|1\rangle$ ($\Phi_{\text{ext}} - 0.24\Phi_0$).

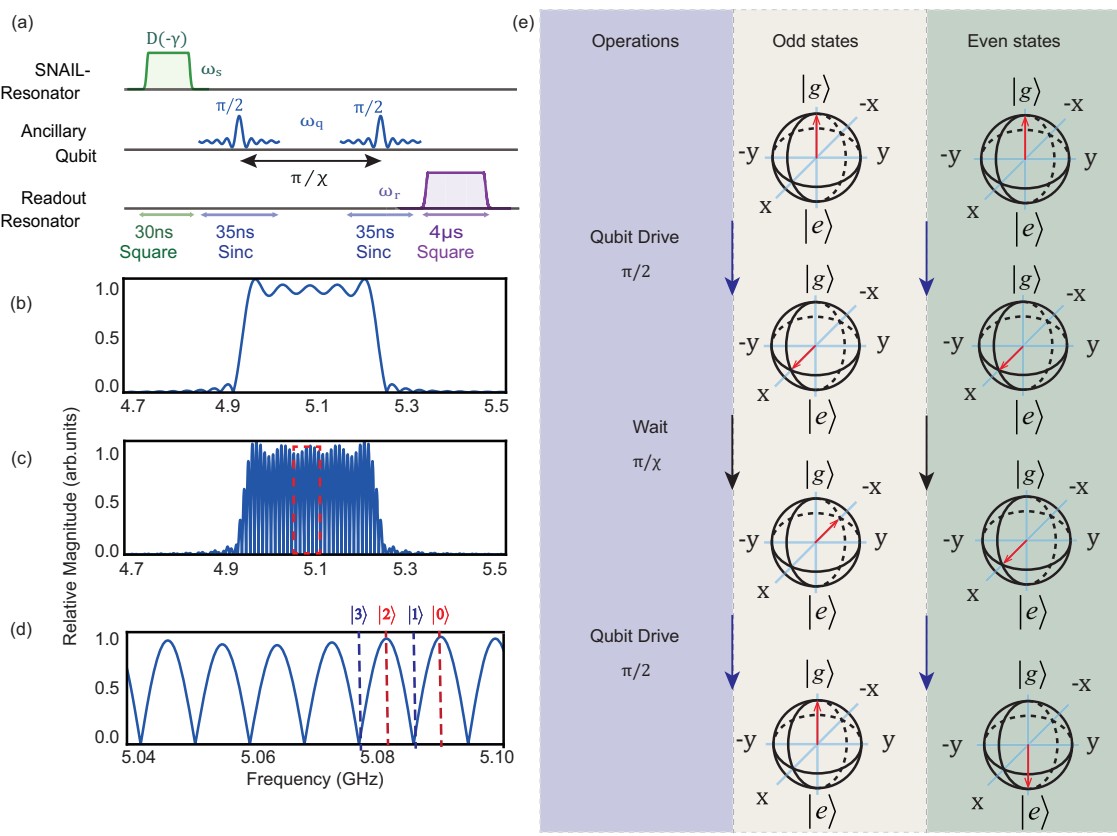

**Fig. 8 | Wigner tomography. a** Pulse sequence for Wigner tomography. **b** Frequency spectrum of a sinc shaped pulse. **c** Frequency spectrum of two sinc pulses with a time spacing $\pi/\chi$. **d** Enlarged view of (**c**). **e** Bloch sphere of the ancillary qubit during the cat state Wigner tomography.

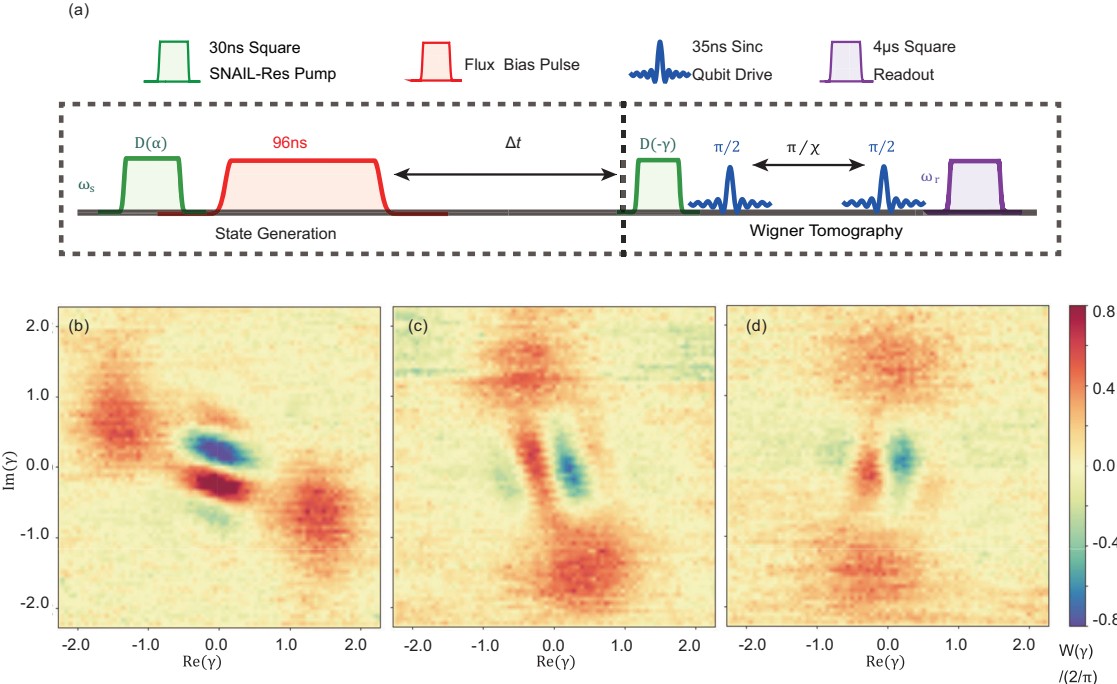

**Fig. 9 | Cat states generation and preservation. a** Pulse sequence for 2-component cat $|\alpha\rangle \pm i|-\alpha\rangle$ preparation. After preparation, the states are measured after $\Delta t =$ (**b**) 0 ns, (**c**) 100 ns, (**d**) 200 ns.

The measurement results from single-tone and two-tone methods agree with the theoretical calculation very well [Fig. 2a].

**Wigner tomography**

The Wigner function of the states is obtained by the parity measurement. For definition, the Wigner function can be described as

$$W(\gamma) = \frac{2}{\pi} Tr[D(-\gamma)\rho D(\gamma)P], \tag{27}$$

$$P = e^{i\pi a^\dagger a} = (-1)^N, \tag{28}$$

By applying a displacement $D(-\gamma)$ to a density operator $\rho$, we get a new state with density operator:

$$\rho' = D(-\gamma)\rho D(\gamma). \tag{29}$$

Thus, the Wigner function $W(\gamma)$ is proportional to the average of parity operator $P$ which can be measured with the sequence in Fig. 8a. Here, we apply two $\pi/2$ pulses to the ancillary qubit. With a time spacing $\pi/\chi$ between two pules, this sequence can be treated as a parity measurement where the state of qubit $|g\rangle$ ($|e\rangle$) corresponds to $P = -1(1)$. The mechanism can be described in the qubit Bloch sphere [Fig. 8e] and signal spectrum [Fig. 8b–d]. The sequence to qubit is $\pi$ ($2\pi$) pulse if photon number $N$ is even (odd). Considering the frequency shift of qubit, we employ a function of $sinc(t) = sin(t)/t$ to cover a larger spectrum range uniformly.

**States preservation**

After the process of states generation, the cat states are preserved by turning the flux bias off. Here, the Wigner functions are measured after a certain evolution time $\Delta t$. As shown in Fig. 9b–d, the Kerr-induced evolution is ideally prevented, the fidelity of 2-component cat ($|\alpha\rangle \pm i|-\alpha\rangle$) state is 89.1%, 81.9% and 75.8% after (0, 100, 200 ns).

## Data availability

The data that support the findings of this study are available in figshare [https://doi.org/10.6084/m9.fgshare.23694369].

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

## Acknowledgements

The authors acknowledge the use of the Nanofabrication Laboratory (NFL) at Chalmers. We wish to express our gratitude to Xiaoming Xie, Lars Jönsson, Fernando Quijandría, Timo Hillmann, and Hang-Xi Li for help. This work is supported in part by the Shanghai Technology Innovation Action Plan Integrated Circuit Technology Support Program (No. 22DZ1100200), the National Natural Science Foundation of China (No. 92065116), Strategic Priority Research Program of the Chinese Academy of Sciences (Grant No. XDA18000000), and the Key-Area Research and Development Program of Guangdong Province, China (No. 2020B0303030002). Y.L. and P.D. acknowledge support from the Knut and Alice Wallenberg Foundation via the Wallenberg Center for Quantum Technology (WACQT) and from the Swedish Research Council (Grant number 2015-00152).

## Author contributions

X.L.H. and Z.R.L. conceived the experiment. X.L.H. performed the experiments and analyzed the data. Y.L. triggered the project, designed, simulated, fabricated the device and helped with the measurement and analysis. D.Q.B. provided the theoretical support. X.L.H. wrote the manuscript together with Y.L. and Z.R.L. H.X., W.B.J., and X.L.H. built up the measurement system. A.F.R. helped with the development of the fabrication recipe. Z.W., P.D., and J.S.T. contributed to discussions of the results. All authors contributed to revising and proofreading of the manuscript. Z.R.L. supervised the project.

## Competing interests

The authors declare no competing interests.
