## [Peer Review File · Nature Communications]

Fast generation of Schrödinger cat states using a Kerr-tunable superconducting resonatorREVIEWER COMMENTS

Reviewer #1 (Remarks to the Author):

This work attempts to demonstrate the generation of Schrodinger cat states in a superconducting resonator. Such states are interesting due to the potential for using them as resource for quantum error-correction. The authors are able to generate cat states in an on-chip SNAIL resonator which gives a tunable Kerr-nonlinearity that can be controlled from zero to a finite value via the flux threaded through the SNAIL loop. The SNAIL-resonator is further part of a larger planar circuit involving an ancillary transmon qubit and readout-resonator, which are used for Wigner tomography. This setup has elements that are borrowed from previous implementations (Ref. 8 and 20) with the new ingredient of fast-flux control of the SNAIL. The description of the experiment and results (Fig. 3, 4) look correct and the fidelities quoted are reasonable for the first iteration of this hardware. As such, the paper is perhaps at the level of other papers published in NPJ Quantum Information but I don't think it is significant enough for it to be published in Nature Communications.

I take serious issue with the authors claim that the cat states are "stabilized" or "passively preserved", which should definitely be revised. This seems to fundamentally misunderstand the meaning of "stabilized" as used by other authors in the field. The authors seem to imply that the cause for the cat states to not be stable is the Kerr-effect and that by turning off the Kerr they can make the cat state stable. However, they do not address the photon loss in the resonator which is the primary reason that cat states will decay to the vacuum. In previous works, where stabilization is claimed (Ref 20, 21), a manifold spanned by cat states is stabilized against photon loss by a combination of continuous driving, Kerr nonlinearity and two-photon dissipation.

The authors are also missing citations to work from the Leghtas group – Nature Physics 16, 509 (2020) and arXiv:2204.09128, which are examples of stabilization of a manifold of cat states in a planar circuit.

Reviewer #2 (Remarks to the Author):

In this paper, the authors report experimental generation of Schrödinger cat states using their novel "SNAIL-terminated resonator." The essential feature of this device is the full controllability of its Kerr nonlinearity from zero to a large value. Harnessing this feature, we can easily generate Schrödinger cat states by generating a coherent state with a microwave drive (displacement operation) followed by switching on Kerr nonlinearity during a finite time. By switching off Kerr nonlinearity after the generation, we can also preserve the generated cat states. The authors have demonstrated all these operations experimentally, the results of which are in good agreement with theoretical ones. In my opinion, the present beautiful experimental results definitely deserve publication in Nature Communications. Before formal acceptance, however, the authors should reconsider the following for improving their manuscript.

To my understanding, the generated cat states are not actively stabilized, but preserved by eliminating Kerr nonlinearity. Also, the essential feature of the present method is not just Kerr-free, but full tunability of Kerr. So in my opinion, the authors should reconsider the title and so on. (For example, "Fast generation and preservation of Schrödinger cat states using a Kerr-tunable superconducting resonator.")

Reviewer #3 (Remarks to the Author):

In their manuscript entitled "Fast generation of stabilized Schrodinger cat states in a Kerr-free superconducting resonator", the authors present a 2D superconducting architecture able to synthesize cat states using a tunable Kerr non-linearity. It comprises a resonator whose frequency is made flux-tunable by including a SNAIL, and whose quantum states can be prepared and measured using an ancillary qubit.

The novelty of the work lies in the use of a SNAIL, which introduce a non-linearity (Kerr effect) for the resonator which is not only tunable but can change signs, providing a flux at which it cancels with the ancillary-qubit induced non-linearity. The existence of this Kerr-free point is essential to prevent the distortion of quantum-states stored into the resonator. This property of the SNAIL is not unknown (<https://arxiv.org/pdf/2212.11929.pdf>) but the authors application of it for providing a Kerr-free flux point for a storage resonator is

interesting. Yet this point has been reported in this article <https://arxiv.org/pdf/2210.09718.pdf>, with one author being a co-author, and with a device that looks identical (or maybe even the same device ?). An explicit reference to this work would be nice when presenting the architecture.

The authors characterize the Kerr effect to be below 70kHz at this Kerr-free point using a spectroscopic scheme to probe the power-induced frequency shift. They then study the evolution of a coherent state at this point, and see no Kerr-induced distortion for 200ns. This is nice, yet should be extended to time $>1/(2\pi \cdot 70\text{kHz}) = 2\mu\text{s}$ for backing the authors claim of a strong cancellation of the Kerr effect.

The authors then use this platform to make a fast preparation of Schrodinger cat states by applying fast biases between this Kerr-free point, and a flux where a significant Kerr effect exist. They achieve 90% fidelity for a 2-legged cat (synthesizing a Y-axis cat state $|\alpha\rangle + i|\alpha\rangle$). This fidelity is poor compared to existing architectures, yet they do so in less than 100ns, which is faster than most techniques for synthesizing cats. Yet considering the fidelity, the speed trade-off seems very costly in the perspective of running a bosonic code.

The authors then claim to have achieved a Z-axis cat state ($|\alpha\rangle - |-\alpha\rangle$) and also "the Y rotation RY and Z rotation RZ and fulfilled the requirement of logic gates.", but I do not see any data backing this statement, even in the methods section. I do not understand how the authors control the initial phase of the cat, and how they can perform gate operation on it. Could the authors provide an explanation ?

Overall, I thus find the platform described by the paper and the achieved results interesting and at the appropriate level for publication in Nature Comm. However, the novelty of the platform is somewhat reduced by the previous publication, and the quality of the proofs somewhat below standard. The paper also misses some key facts and discussions and has a few misleading/unclear statements.

Title: The authors call their manuscript "Fast generation of stabilized Schrodinger cat states in a Kerr-free superconducting resonator" which seem to me misleading : the resonator only

has one flux-point where it is linear, and the authors make use of its non-linearity for the cat generation.

Abstract & I 29-30 and near the conclusion: The authors state repeatedly that cat states are typically generated in 3D architectures, yet state of the art results have also been achieved in 2D (for example <https://doi.org/10.1038/s41567-020-0824-x>), so I do not think this distinction is warranted.

I 46: "It allows us to prepare cat states." This statement is rather obscure, and should be backed by references.

I 47-48: the authors claim: "We can also preserve the prepared cat states, because the Kerr-induced evolution is prevented." yet they do not provide any timescale measurement - except for fig 3 which only extends to 200ns, and is for coherent states !

Fig 2c: displaying the fitted frequencies would be helpful

No figures are given for the T1, T2 times of the qubit, and the linewidth of the SNAIL resonator is also not given. It would be nice to give them, especially since I expect these figures are somewhat less than what achieved in 3D architecture, and thus should be taken into account in the 3D/2D discussion.

What actually limits the fidelity of the cat state preparation ? Qutip simulation are given, but not analyzed.

In the method, Fig 9 shows the evolution of the cat state after its preparation. I find this data would deserve to be included in the main text. It looks like the cat decay in about ~200ns. Could the authors explain where does this limitation come from ?

We thank the referees for their careful reading and positive evaluations to the manuscript. For addressing each referee's specific comments, here we list all the comments from the referees and our point-by-point response. The corresponding response and supplements to the comments have been synchronized to the revised manuscript.

Response to reviewer #1:

- (1) This work attempts to demonstrate the generation of Schrodinger cat states in a superconducting resonator. Such states are interesting due to the potential for using them as resource for quantum error-correction. The authors are able to generate cat states in an on-chip SNAIL resonator which gives a tunable Kerr-nonlinearity that can be controlled from zero to a finite value via the flux threaded through the SNAIL loop. The SNAIL-resonator is further part of a larger planar circuit involving an ancillary transmon qubit and readout-resonator, which are used for Wigner tomography. This setup has elements that are borrowed from previous implementations (Ref. 8 and 20) with the new ingredient of fast-flux control of the SNAIL. The description of the experiment and results (Fig. 3, 4) look correct and the fidelities quoted are reasonable for the first iteration of this hardware. As such, the paper is perhaps at the level of other papers published in NPJ Quantum Information but I don't think it is significant enough for it to be published in Nature Communications.

Re: We thank the referee for the positive evaluations and recognition of the novelty in our work. As pointed out by the reviewer, the innovation in our work is that we report a novel strategy to generate and preserve of Schrödinger cat states in an on-chip nonlinear superconducting resonator by fast Kerr nonlinearity modulation. By using the fast Kerr nonlinearity modulation, we generate cat states by fast Kerr interaction induced evolution and preserve them in a Kerr-free linear mode by tuning the device to a work point where the four-wave mixing coupling is suppressed. This scheme is made possible by the resonator's inherent nonlinearity, which enables universal control of bosonic modes stored inside. In addition to the high scalability, our circuits possess several noteworthy advantages. We acknowledge that our approach in utilizing Wigner tomography is indeed inspired by previous methodologies, which have now become widely adopted. However, it is important to note that our main objective is to demonstrate a novel approach for preparing Schrödinger cat states. We firmly believe that this does not diminish the innovative contribution of our manuscript. We are confident that our paper meets the rigorous requirements and standards established by Nature Communications.

- (2) I take serious issue with the authors claim that the cat states are “stabilized” or “passively preserved”, which should definitely be revised. This seems to fundamentally misunderstand the meaning of “stabilized” as used by other authors in the field. The authors seem to imply that the cause for the cat states to not be stable is the Kerr-effect and that by turning off the Kerr they can make the cat state stable.....In previous works, where stabilization is claimed (Ref 20, 21), a manifold spanned by cat states is stabilized against photon loss by a combination of continuous driving, Kerr nonlinearity and two-photon dissipation.

Re: Regarding the title of the manuscript, we were emphasizing no further nonlinear evolution would damage the photon states after switching the system back to the linear work point. The meaning of ‘stabilization’ here is not the same as what was reported in the previous papers (Nature, 584, pages205–209 (2020)), in which the states lay in the potential valleys. Our strategy was based on the Kerr nonlinearity with a large dynamic range (from -4 to +5MHz), the cat states were generated and protected by Kerr tuning. Thus, to avoid any misunderstanding comparing to the previous publications, we would change the title to ‘Fast generation and preservation of Schrödinger cat states in a Kerr-tunable superconducting resonator’. Throughout the manuscript, we have used the term "preservation" instead of "stabilization."

(3) However, they do not address the photon loss in the resonator which is the primary reason that cat states will decay to the vacuum.

Re: With our structure, the cross Kerr nonlinearity can be ideally compensated. As a result, the cat states can be passively stored without any microwave pumping. We agree that photon loss is still the dominated source of collapse at the moment. We did a simulation under a lossy Hamiltonian.

With a single photon loss rate $\kappa/2\pi=200\text{kHz}$, the interference fringe of Wigner function disappeared within several hundreds of nanoseconds. However, a small Kerr coefficient (e.g. 50kHz, which is a normal value of a cross Kerr term) also affect the state within the same time. We could mostly overcome the single photon loss by many previous strategies (npj Quantum Information, volume 8, article number: 3 (2022)). Here, we achieved fully control on the Kerr coefficient. We utilized the Kerr nonlinearity to prepare the states and handle the unwanted effects. Meanwhile, compared to engineering the two-photon dissipation via the adiabatic process, the generation of our Schrödinger cat states is several times faster. (Nature, volume 584, pages205–209 (2020), npj Quantum Information, volume 3, Article number: 18 (2017)).

To improve the performance of our state operations, further study on the quality factor of the SNAIL-resonator is necessary. Once the quality factor become $1e6$ or higher (a typical value of a high quality CPW resonator), the fidelity of the 2-legged cat preparation may reach 99.8%.

(4) The authors are also missing citations to work from the Leghtas group – Nature Physics 16, 509 (2020) and arXiv:2204.09128, which are examples of stabilization of a manifold of cat states in a planar circuit.

Re: The publications from Prof. Leghtas group (Nature Physics 16, 509 (2020) and arXiv:2204.09128) are cited in the discussion and conclusion parts. We agree it is better to make a comparison at the beginning of the paper. Then, we have added on sentence at the end of the first paragraph in Introduction.

To reviewer #2

(1) In this paper, the authors report experimental generation of Schrödinger cat states using their novel "SNAIL-terminated resonator." The essential feature of this device is the full controllability of its Kerr nonlinearity from zero to a large value. Harnessing this feature, we can easily generate Schrödinger cat states by generating a coherent state with a microwave drive (displacement operation) followed by switching on Kerr nonlinearity during a finite time. By switching off Kerr nonlinearity after the generation, we can also preserve the generated cat states. The authors have demonstrated all these operations experimentally, the results of which are in good agreement with theoretical ones. In my opinion, the present beautiful experimental results definitely deserve publication in Nature Communications.

Re: We sincerely appreciate the referee's valuable opinion, and we are grateful for the constructive comments provided below. We share the sentiment and strongly believe that Nature Communications is an ideal platform to showcase our findings to a broad scientific audience.

(2) Before formal acceptance, however, the authors should reconsider the following for improving their manuscript. To my understanding, the generated cat states are not actively stabilized, but preserved by eliminating Kerr nonlinearity. Also, the essential feature of the present method is not just Kerr-free, but full tunability of Kerr. So in my opinion, the authors should reconsider the title and so on. (For example, "Fast generation and preservation of Schrödinger cat states using a Kerr-tunable superconducting resonator.")

Re: Regarding the title of the manuscript, we were emphasizing no further nonlinear evolution would damage the photon states after switching the system back to the linear work point. The meaning of 'stabilization' here is not the same as what was reported in the previous papers (Nature, 584, pages205–209 (2020), Scientific Reports, 13, Article number: 1606 (2023)), in which the states lay in the potential valleys. Our strategy was based on the Kerr nonlinearity with a large dynamic range (from -4 to +5MHz), the cat states were generated and protected by Kerr tuning. Thus, to avoid any misunderstanding comparing to the previous publications, we think the title suggested by the referee is suitable for our manuscript, and then accept it,

To reviewer #3

- (1) In their manuscript entitled "Fast generation of stabilized Schrodinger cat states in a Kerr-free superconducting resonator", the authors present a 2D superconducting architecture able to synthesize cat states using a tunable Kerr non-linearity. It comprises a resonator whose frequency is made flux-tunable by including a SNAIL, and whose quantum states can be prepared and measured using an ancillary qubit.

Re: We thank the referee for the careful reading and insightful comments. Our main innovation lies in the invention of a novel strategy to generate and stabilize Schrödinger cat states in a nonlinear superconducting resonator by using a fast tunable Kerr non-linearity. The quantum states can be prepared by the nonlinear resonator's inherent nonlinearity and then measured using an ancillary qubit as pointed out by the referee.

- (2) The novelty of the work lies in the use of a SNAIL, which introduce a non-linearity (Kerr effect) for the resonator which is not only tunable but can change signs, providing a flux at which it cancels with the ancillary-qubit induced non-linearity. The existence of this Kerr-free point is essential to prevent the distortion of quantum-states stored into the resonator. This property of the SNAIL is not unknown (<https://arxiv.org/pdf/2212.11929.pdf>), but the authors application of it for providing a Kerr-free flux point for a storage resonator is interesting. Yet this point has been reported in this article <https://arxiv.org/pdf/2210.09718.pdf>, with one author being a co-author, and with a device that looks identical (or maybe even the same device ?). An explicit reference to this work would be nice when presenting the architecture.

Re: Yes, the device was originally designed and fabricated by Dr. Yong Lu and Prof. Delsing from Chalmers. A citation has been added while discussing the device structure.

- (3) The authors characterize the Kerr effect to be below 70kHz at this Kerr-free point using a spectroscopic scheme to probe the power-induced frequency shift. They then study the evolution of a coherent state at this point, and see no Kerr-induced distortion for 200ns. This is nice, yet should be extended to time $>1/(2\pi \cdot 70\text{kHz}) = 2\mu\text{s}$ for backing the authors claim of a strong cancellation of the Kerr effect.

Re: Near the Kerr-free work spot, we used 1-tone measurement to measure the Kerr nonlinearity of the SNAIL-terminated resonator. A 500 ns Gauss-shaped pulse was used to inject photons in this measurement, as shown in Fig. 2. Frequency shift was not observable at the working point with the photon number $N=0-16$ ($\alpha=0-4$) in Fig. 2(d). We thus estimated the Kerr nonlinearity $|K/2\pi| < \Delta f/N \sim 70\text{kHz}$ (Δf (~ 1 MHz) is the half line width of the photon injection pulse) due to the resolution limited by the pulse width (500 ns).

It is theoretically possible to verify the small Kerr nonlinearity, either by a longer photon injection pulse or tracking the Kerr-induced evolution under a longer period. However, as shown in the supplementary material (Fig. S2), the decoherence time T_2 for the photon in the SNAIL terminated

resonator is measured through a Ramsey-like sequence. The T_2 is around $1\mu\text{s}$ for single photon and even worse for larger photon number. The measurement with a long ($>1\mu\text{s}$) pulse sequence is affected by the decoherence of the light field.

Here, we prepared a coherent state with $\alpha \sim 2$ and measured the Wigner function after hundreds (0,100,200,500) ns evolution, as what we described in Fig. 3(a). Additionally, we calculate the Wigner function under Kerr Hamiltonian:

$$H_K = K a^\dagger a^\dagger a a$$

From the experimental results, it is clear that the interference fringes disappear after 500 ns, which means the dephasing process affects in a long-time experiment. Therefore, due to the limitation of our device, we are regret to say that the measurement suggested by the referee can not be performed at the moment whereas it will be definitely possible after improving the device quality.

(4) The authors then use this platform to make a fast preparation of Schrodinger cat states by applying fast biases between this Kerr-free point, and a flux where a significant Kerr effect exist. They achieve 90% fidelity for a 2-legged cat (synthesizing a Y-axis cat state $|\alpha\rangle + i|\alpha\rangle$). This fidelity is poor compared to existing architectures, yet they do so in less than 100ns, which is faster than most techniques for synthesizing cats. Yet considering the fidelity, the speed trade-off seems very costly in the perspective of running a bosonic code.

Re: The fidelity we measured in the experiments is now affected by the decoherence and relaxation of the photons, not limited by the speed. By optimizing the design parameters, we can improve both the decoherence time and the speed. The fidelity of 99% is achievable with a 40ns gate and 5us decoherence time.

Regarding the speed of the $|\alpha\rangle$ to $|\alpha \pm i\rangle$ gate, further optimization on Kerr coefficient may benefit both the speed and fidelity. For example, it is possible to set the Kerr-free work point very close to the frequency sweet point.

Here, we provide some simulation results about a possible design of SNAIL-terminated resonator. The dynamic range of Kerr coefficient (around -14MHz~+500kHz) makes it possible to compensate the cross Kerr term. Thus, the flux bias of Kerr-free work point is very close to $0.5\Phi_0$ where the resonant frequency reaches the minimum value. By lowering the gradient $|df/d\Phi|$, we expect a longer decoherence time.

Additionally, with the parameters shown above, the maximum nonlinearity $|K|$ is also larger (>14 MHz). As we discussed in the manuscript, the gate time of $|\alpha\rangle$ to $|\alpha \pm i\rangle$ operation is proportional to $1/|K|$. The gate time become less than 40ns with the optimized design. Theoretically, it is also possible to achieve a larger nonlinearity, but we believe 40ns is a proper value under the microseconds coherent time of the quantum circuit and the normal speed of the control electronics.

(5) The authors then claim to have achieved a Z-axis cat state ($|\alpha\rangle-|-\alpha\rangle$) and also "the Y rotation RY and Z rotation RZ and fulfilled the requirement of logic gates.", but I do not see any data backing this statement, even in the methods section. I do not understand how the authors control the initial phase of the cat, and how they can perform gate operation on it. Could the authors provide an explanation?

Re: To fulfill the requirements of a cat qubit (see the ref [23]: Nature, volume 584, pages205–209 (2020)), the relative phase between the two coherent states is supposed to be controllable for a two-legged cat states.

With the strategy we reported in the main body of the manuscript, we achieved the gate between a coherent state $|\alpha\rangle$ and a superposition $|\alpha \pm i\rangle$.

The other axis ($|\alpha \pm i\rangle$) can be achieved by taking the ancillary qubit in use. The phase difference of the excited and ground states of the ancillary qubit can be transferred to the coherent states ($|\alpha\rangle$ and $|-\alpha\rangle$). Recently, this method was used built up universal control of photon states (Echoed Conditional Displacement (ECD)(Nature Physics, volume 18, pages1464–1469 (2022)), Selective Number-dependent Arbitrary Phase (SNAP)(Physical Review A 92, 040303(R) (2015))). The initialization operations we reported in the supplementary materials can be described in the following table (odd parity).

Step	Operations	Target States ($ Qubit, Resonator\rangle$)
1	Wait for relaxation	$ g, 0\rangle$
2	$\pi/2$ on Qubit	$1/\sqrt{2} e, 0\rangle + 1/\sqrt{2} g, 0\rangle$
3	Displace α on resonator	$1/\sqrt{2} e, \alpha\rangle + 1/\sqrt{2} g, \alpha\rangle$
4	Wait for $t=1/2\chi$	$1/\sqrt{2} e, -\alpha\rangle - 1/\sqrt{2} g, \alpha\rangle$
5	Displace α on resonator	$1/\sqrt{2} e, 0\rangle - 1/\sqrt{2} g, 2\alpha\rangle$
6	Conditional π on Qubit (Available for small photon number)	$1/\sqrt{2} g, 0\rangle - 1/\sqrt{2} g, 2\alpha\rangle$
7	Displace $-\alpha$ on resonator	$1/\sqrt{2} g, -\alpha\rangle - 1/\sqrt{2} g, \alpha\rangle$

The Wigner functions of the photons after each steps are shown below:

Notice that in step 4 and 5, the target states are maximally entangled states. The interference fringes are therefore disappeared. From step 5 to step 7, we try to achieve a reset operation on the ancillary qubit, which can be expressed as:

$$|g\rangle\langle g| + e^{i\theta}|g\rangle\langle e|$$

The phase difference of the two coherent states depends on the value of θ .

We can achieve the transition between $|\alpha\rangle$ and $|\alpha\rangle + \exp(i\varphi)|-\alpha\rangle$ with this method. The pulse sequence and measured Wigner function are shown in Fig. S1, where we can clearly see the

interference fringes of odd parity. With this method, the pulse width of the conditional π pulse in step 5 is limited by the dispersive shift ($t \sim 2\pi/\chi$, ~ 500 ns in our experiment). Moreover, a part of the photon states is displaced to $|2\alpha\rangle$ during step 5, the decoherence/loss is enhanced by the large photon number. The fidelity of our experimental result (fig S1) is therefore affected a lot by the time consumption in step 5.

(6) Overall, I thus find the platform described by the paper and the achieved results interesting and at the appropriate level for publication in Nature Comm. However, the novelty of the platform is somewhat reduced by the previous publication, and the quality of the proofs somewhat below standard. The paper also misses some key facts and discussions and has a few misleading/unclear statements.

Re: We thank the referee for the positive evaluations and the comments raised by the referee are constructive to make the manuscript more rigorous. Regarding the novelty of the platform, we understand the referee's perspective that it may be somewhat reduced due to a previous publication where the publication is only for characterizing the device by our coauthors. Very differently, by using a similar device, our manuscript reports a novel strategy to generate and preserve Schrödinger cat states in a nonlinear superconducting resonator by fast Kerr nonlinearity modulation, which represents a novel method and a new approach. We believe that our experimental results and findings offer valuable insights for publication in Nature Communications.

(7) Title: The authors call their manuscript "Fast generation of stabilized Schrodinger cat states in a Kerr-free superconducting resonator" which seem to me misleading : the resonator only has one flux-point where it is linear, and the authors make use of its non-linearity for the cat generation.

Re: Regarding the title of the manuscript, we were emphasizing no further nonlinear evolution would damage the photon states after switching the system back to the linear work point. The meaning of 'stabilization' here is not the same as what was reported in the previous papers (Nature volume 584, pages205–209 (2020)), in which the states lay in the potential valleys. Our strategy was based on the Kerr nonlinearity with a large tuning range (from -4 to +5MHz), the cat states were generated and protected by fast Kerr interaction switch. Thus, to avoid any misunderstanding comparing to the previous publications, we would change the title to "Fast generation and preservation of Schrödinger cat states in a Kerr-tunable superconducting resonator".

(8) Abstract & I 29-30 and near the conclusion: The authors state repeatedly that cat states are typically generated in 3D architectures, yet state of the art results have also been achieved in 2D (for example <https://doi.org/10.1038/s41567-020-0824-x>), so I do not think this distinction is warranted.

Re: We understand that there are different ways to achieve scalable Bosonic-code based quantum information processors. Several groups are investigating the photon states in 2D structures (Nature Physics volume 16, pages509–513 (2020), arXiv:2204.09128, Phys. Rev. A 105, 023519). What we

want to expressed is the 3D structure is more common and reliable for bosonic codes (Nature volume 616, pages56–60 (2023), Nature volume 616, pages50–55 (2023)). The sentences in the manuscript are revised to be more exact.

E.g.

Line 3, Page 1 in abstract, 'Especially, cat states in a phase space protected against phase-flip errors can be used as a logical qubit. However, cat states, normally generated in three-dimensional cavities and/or strong multi-photon drives, are facing the challenges of scalability and controllability.'

Line 20, Page 2 in main text, "In previous results, Schrödinger's cat states were mostly generated by engineering the two-photon losses or ancilla-assisted processes in two- and three- dimensional structures.'

(9) 1 46: "It allows us to prepare cat states." This statement is rather obscure, and should be backed by references.

Re: Some statements in the manuscripts have been revised.

E.g.

'The fast tunable nonlinearity is utilized to prepare cat states in our strategy.'

(10) 1 47-48: the authors claim: "We can also preserve the prepared cat states, because the Kerr-induced evolution is prevented." yet they do not provide any timescale measurement - except for fig 3 which only extends to 200ns, and is for coherent states ! " and "In the method, Fig 9 shows the evolution of the cat state after its preparation. I find this data would deserve to be included in the main text. It looks like the cat decay in about ~200ns. Could the authors explain where does this limitation come from?"

Re: The evolution caused by Kerr nonlinearity can be mostly suppressed near the Kerr-free work point. As we showed in Fig. 3 and Fig. 9, the relative phases among Fock states are stable after the state preparation.

However, the decoherence time of the photons in the SNAIL-terminated resonator is not long enough in our device. As what we showed in the supplementary materials (Fig. S2), the decoherence time T_2 is around 1us for a small photon number and even worse for a larger photon number.

From theoretical calculation (npj Quantum Information volume 3, Article number: 18 (2017)), the single photon loss can be treated as a source of dephasing for a cat state. The following figures show the Wigner function for a 2-legged cat state in a system with a small Kerr nonlinearity ($K/2\pi=50\text{kHz}$) and a single photon loss rate ($\kappa/2\pi=200\text{kHz}$), which is similar with our experimental condition in Fig. 9.

Although the life time of a single photon is several microseconds, the interference fringes disappear within hundreds of nanoseconds. Also, other mechanisms, such as two photon loss or flux noise, may damage the photon states. The results will be improved in future experiments.

In our manuscript, we extensively discuss the primary reasons and limitations associated with the decay of cat states (1st paragraph in Page. 7), as well as propose measures for improvement.

The strategy to organize the paper is to find the Kerr free point, verified further by a simple measurement, namely, the evolution of the coherence state, as a first step. Afterwards, we prepare a cat state at the Kerr-free point. To observe the Kerr effect on the state in the resonator, a measurement taken on coherent states is more straightforward and simpler.

(11) Fig 2c: displaying the fitted frequencies would be helpful

Re: The figure has been modified in the revised manuscript. We used a 500 ns Gauss-shaped (with the standard error=500ns/6) pulse to inject photon in the experiments described in Fig.2 (c,d,e). As a result, the fitting of the resonant frequency is not very accurate. Indicators are added in the figures.

(12) No figures are given for the T1, T2 times of the qubit, and the linewidth of the SNAIL resonator is also not given. It would be nice to give them, especially since I expect these figures are somewhat less than what achieved in 3D architecture, and thus should be taken into account in the 3D/2D discussion.

Re: Thank you for clarifying. The T1 and T2 of the ancillary qubit are shown below. In our structure, the relaxation and decoherence rate are affected by the frequency of the coupled nonlinear resonator. Overall, the T1 and T2 of the qubit stay at the level of 10 μ s. The revised supplementary materials now include the presentation of the data as mentioned.

By simultaneously injecting photons and driving the qubit through the charging line, the linewidth

of the SNAIL-terminated resonator can be measured with CW signals. The results show that the FWHM is 377 kHz under -140 dBm power. (These results have been added in the supplementary Fig. S3)

(13) What actually limits the fidelity of the cat state preparation? Qutip simulation are given, but not analyzed.

In our experiments, the fidelity is mainly affected by the decoherence time (~ 1 microsecond) of the SNAIL-terminated resonator. Since the duration of a 2-legged cat preparation is 96 ns in current experiment, the highest fidelity is approximated to be 91%.

The possible decoherence source of the SNAIL-Resonator may include:

- a. The flux noise through the SNAIL loop.
- b. Coupled ancillary qubit.
- c. Internal single/two photon loss.

We think photon loss is the dominated source of collapse at the moment. As we discussed above, 200kHz single photon loss may totally destroy the cat states with in hundreds nanoseconds.

To improve the performance of our state operations, further study on the quality factor of the SNAIL-resonator is necessary. Once the quality factor become $1e6$ or higher (a typical value of a high quality CPW resonator), the fidelity of the 2-legged cat preparation may reach 99.8%. Some details have been added in the conclusion part and supplementary.

REVIEWERS' COMMENTS

Reviewer #1 (Remarks to the Author):

The authors have made some improvements in the manuscript, especially in clarifying that the Schrodinger cat states are not actually stabilized as claimed in the previous version and pointed out by all referees. The new title with the word “preservation” is somewhat better than “stabilization” though I personally think it is still too close and is liable to be misunderstood by the readers without more clarification. My personal title would just be “Fast generation of Schrodinger cat states using a Kerr-tunable superconducting resonator”. The authors are essentially doing a unitary pulse sequence consisting of a displacement followed by a Kerr gate. The innovation is that the Kerr gate in the authors device is a true gate which one can turn on/off unlike Ref. 21. The authors should clearly say in the introduction and conclusion that they are preserving the cat states against Kerr-induced evolution (a coherent error) rather than photon loss (an incoherent error).

The authors sentences in line#30-40 makes it seem like the previous cat state stabilization schemes are the primary past work they are improving upon. But actually, their state preparation scheme is an alternate to traditional gate-based cavity control schemes using the dispersive shift of a nominally linear resonator to an ancilla qubit such as QCMAP (Ref. 8), SNAP (Phys. Rev. Lett. 115, 137002), GRAPE (Nature Communications volume 8, Article number: 94) and ECD (Ref. 24). It is definitely possible to design superconducting resonators with low Kerr but still make cat states quickly using the dispersive coupling to an ancilla qubit. If the goal is just to show state preparation via coherent control, the authors should make more clear comparisons with these existing approaches for cat state preparation in 2D and 3D cavities (I don't think this distinction is germane but the authors are bringing it up). In particular the above approaches have a typical timescale of $1/\text{dispersive shift}$, so given $\chi/2\pi \sim 4$ MHz (similar to the authors ancilla) it would be about 250 ns to prepare a cat state. With the authors approach this time is reduced to about 100 ns. However, the T_1 , T_2 times of the resonator are significantly shorter than previous implementations so it is not clear that the faster state preparation would actually lead to higher fidelity. There are also good reasons to believe that a SNAIL resonator would always have lower T_1 compared to a linear CPW resonator since there are many extra fab steps and higher participation of lossy

surfaces in a SNAIL resonator.

Overall, this work is an interesting new method to realize cat states of a superconducting resonator and is definitely at the level of NPJQI. I think there are still unknowns as to whether having a nonlinear resonator based on a SNAIL is better approach than using the dispersive coupling to the ancilla (which is anyway present for tomography/readout etc), especially given the worse coherence of this type of hardware over a purely linear cavity. However, if the other referees continue to feel that this paper rises to the level of Nature Communications, I would be also willing to recommend it.

Reviewer #3 (Remarks to the Author):

I thank the authors for their careful and precise answers to my comments. I agree with the title change, and the modifications the authors made to the manuscript.

From the authors' answer I now realize that proper assessment of their device, specifically accurately characterizing the Kerr-freeness, is limited by their device performance, namely in the short coherence time of their Kerr-tunable resonator. It is still unclear to me whether this is an intrinsic fault of the design, or if that is something that can be improved with better layout/fabrication techniques to one day be able to use this technique for bosonic encoding. Nevertheless, the authors make a good job of demonstrating fast generation of cat states despite these limited performances, and I think the demonstration is worthwhile publishing in Nature Comm as is.

I however still take exception to the authors claim of :

"In

242 summary, we have achieved the Y rotation RY
243 and Z rotation RZ and fulfilled the requirement of
244 logic gates. It indicates that our platform can be
245 regarded as a logical qubit."

In my opinion, achieving Ry and RZ logic gates require demonstrating that any state on the

Bloch sphere is undergoing the appropriate rotation, for instance by using quantum state tomography. At minima, the authors should show Rabi oscillations on the cat encoding to be able to give such a statement.

I would thus either remove the statement, or just indicate it as a possibility for future work.

We are grateful to the referees who have taken their time to go through our manuscript in detail. We are happy to see that both the referees are positive about our work. We have answered their comments carefully below (for clarity, our responses are marked in black).

To Reviewer #1

The new title with the word “preservation” is somewhat better than “stabilization” though I personally think it is still too close and is liable to be misunderstood by the readers without more clarification. My personal title would just be “Fast generation of Schrodinger cat states using a Kerr-tunable superconducting resonator”. The authors are essentially doing a unitary pulse sequence consisting of a displacement followed by a Kerr gate.

Re:

We agree with the referee suggestion so that we have modified our title as the referee suggested.

The authors should clearly say in the introduction and conclusion that they are preserving the cat states against Kerr-induced evolution (a coherent error) rather than photon loss (an incoherent error).

Re:

According to the reviewer’s suggestions, we modified some sentences in the second paragraph of Introduction and the second last paragraph of conclusion.

The authors sentences in line#30-40 makes it seem like the previous cat state stabilization schemes are the primary past work they are improving upon. But actually, their state preparation scheme is an alternate to traditional gate-based cavity control schemes using the dispersive shift of a nominally linear resonator to an ancilla qubit such as QCMAP (Ref. 8), SNAP (Phys. Rev. Lett. 115, 137002), GRAPE (Nature Communications volume 8, Article number: 94) and ECD (Ref. 24).

Re:

According to the feedback of the reviewer, we revised the second paragraph of Introduction. Hopefully, the sentences are more clear.

It is definitely possible to design superconducting resonators with low Kerr but still make cat states quickly using the dispersive coupling to an ancilla qubit. If the goal is just to show state preparation via coherent control, the authors should make more clear comparisons with these existing approaches for cat state preparation in 2D and 3D cavities (I don’t think this distinction is germane but the authors are bringing it up). In particular the above approaches have a typical timescale of $1/\text{dispersive shift}$, so given $\chi/2\pi \sim 4$ MHz (similar to the authors ancilla) it would be about 250 ns to prepare a cat state. With

the authors approach this time is reduced to about 100 ns. However, the T1, T2 times of the resonator are significantly shorter than previous implementations so it is not clear that the faster state preparation would actually lead to higher fidelity. There are also good reasons to believe that a SNAIL resonator would always have lower T1 compared to a linear CPW resonator since there are many extra fab steps and higher participation of lossy surfaces in a SNAIL resonator.

Re:

Using a dispersively coupled qubit, the gate time is limited by the dispersive shift (normally at the level of MHz). It is therefore possible to generate cat states within hundreds nanoseconds. However, a too strong coupling between qubit and resonator may induce some unwanted effects. In our case, the speed is proportional to the max Kerr coefficient ($K/2 \pi \sim 6$ MHz in this device). By an optimized design, the value of Kerr coefficient can be extremely larger. It is practical to generate cat states within tens nanoseconds. (We have demonstrated a 48ns gate to prepare a 4-legged cat state with our present device.) We believe our approach owns the advantage of speed. Moreover, a strong coupled qubit can be treated as a loss channel of the information encoded in a bosonic system. We intend to minimize the participation of the ancillary qubit during the state preparation. The fidelity of the qubit gates can barely affect the state generation in our method.

Regarding the structure of the device, we agree that the physics behind is almost the same for a two- or three-dimensional resonator. With the state-of-the-art results, a three-dimensional resonator has a larger quality factor, but a two-dimensional structure on chip is a better platform for larger-scale circuits.

By introducing a Josephson Junction loop, the T1 and T2 of the resonator become worse. It is understandable that the relaxation and dephasing processes are enhanced by the non-ideal interfaces and fabrication steps. Same as transmon qubits, the performance can be improved by new materials or carefully designed structures. After several iterations, we believe the lifetime would be similar with transmon qubits'.

To Reviewer #3

From the authors' answer I now realize that proper assessment of their device, specifically accurately characterizing the Kerr-freeness, is limited by their device performance, namely in the short coherence time of their Kerr-tunable resonator. It is still unclear to me whether this is an intrinsic fault of the design, or if that is something that can be improved with better layout/fabrication techniques to one day be able to use this technique for bosonic encoding.

Re:

Comparing with a standard linear resonator, the T1 and T2 of the SNAIL resonator become worse because of the additional Josephson Junction loop. It is understandable that the relaxation and dephasing processes are enhanced by the non-ideal interfaces and fabrication steps. Same as transmon qubits, the performance can be improved by new materials or carefully designed structures. Moreover, the device is also limited by other losses, very possibly from chip mode which can be suppressed by changing the design of the sample box. After several iterations, we believe the lifetime would be similar with transmon qubits'.

I however still take exception to the authors claim of :

"In

242 summary, we have achieved the Y rotation RY
243 and Z rotation RZ and fulfilled the requirement of
244 logic gates. It indicates that our platform can be
245 regarded as a logical qubit."

In my opinion, achieving Ry and RZ logic gates require demonstrating that any state on the Bloch sphere is undergoing the appropriate rotation, for instance by using quantum state tomography. At minima, the authors should show Rabi oscillations on the cat encoding to be able to give such a statement.

I would thus either remove the statement, or just indicate it as a possibility for future work.

Re:

We understand the strategy we described here is not the transversal operation on the Bloch sphere. We therefore modified these sentences in the third last paragraph of Results based on the reviewer's suggestion.